# Cultural Tightness-Looseness: How Was It Related to Vaccination Behavior among University Students in Japan, the United States, and India?

**DOI:** 10.3390/vaccines11121821

**Published:** 2023-12-05

**Authors:** Yoko Kawamura, Mio Kato, Rina Miyawaki, Hirono Ishikawa, Jessica Legge Muilenburg, Yuki Azaad Tomar

**Affiliations:** 1School of Health Sciences, University of Occupational and Environmental Health, Kitakyushu 807-8555, Japan; 2Center for Emergency Preparedness and Response, National Institute of Infectious Diseases, Shinjuku-ku 162-8640, Japan; 3School of Arts and Letters, Meiji University, Suginami-ku 168-8555, Japan; 4Graduate School of Public Health, Teikyo University, Itabashi-ku 173-8605, Japan; 5College of Public Health, University of Georgia, Athens, GA 30602, USA; 6Institute of Home Economics, University of Delhi, Delhi 110016, India

**Keywords:** COVID-19, vaccination, cultural tightness-looseness, behavior, perceptions, international comparison, university students

## Abstract

As a next step to better understand the role of cultural tightness-looseness (CTL), this study aimed to examine whether CTL is associated with COVID-19 vaccination behavior among university students, taking into consideration sociocultural perceptions of vaccination across countries. A global online survey was conducted. University students from Japan, the US, and India participated. The average CTL score, three sociocultural perceptions related to COVID-19 vaccination, side effects, infection experience of themselves and family members, and other demographic variables were used to identify the model and to explain the second vaccination status using stepwise logistic regression methods with Akaike Information Criterion (AIC) scores which was for both the total, with the country as a variable, and for each country. Analyses of data from 1289 respondents who received the first vaccine revealed the essential role of CTL in individuals getting the second vaccine, while also revealing differences between countries. Regardless of the limitations, this study adds knowledge about CTL’s roles in the COVID-19 vaccination behavior among young generations and provides insights into public health communication practices for issues like COVID-19.

## 1. Introduction

### 1.1. The Context of COVID-19 Vaccination

In early 2023, the COVID-19 pandemic was globally marked by a significant reduction in hospitalization rates, ICU admissions, and deaths across all age groups. The population seroprevalence levels of the SARS-CoV2 virus, which causes COVID-19 (Coronavirus Disease 2019), are assumed to be above 90% in most countries, reflecting the combined exposure to infection, vaccination, or both [1]. Among these factors, increasing population-level immunity due to vaccinations has been one of the most important in controlling the disease. 

Considering this situation, the World Health Organization (WHO) has updated the global guidance for vaccination strategies to include the rates of high population immunity, declining risk of mortality and severe disease, differential vaccine performance against infection, and severe disease outcomes. It does not recommend additional routine boosters for medium-risk groups [2]. 

To reach this point, however, the challenges in getting as many individuals as possible vaccinated were many—from technical, infrastructural, social, and political to sociopsychological.

How vaccination progressed differed by the context in which people shared time, physical and social resources, and perceptions. COVID-19 was a unique global challenge and different from other vaccine-recommended diseases such as influenza and Human Papillomavirus (HPV). We should learn what actions we should take to promote behaviors such as vaccination among the youth under an emergency situation like the COVID-19 pandemic. Such actions will differ among different cultural contexts, and comparing countries with different cultures will allow us to identify them.

### 1.2. Youth and University Students

Youth and young adults or university students are a unique population because they are one of the most information-technologically advanced groups, having grown up with smartphones, tablets, PCs, and other gadgets and being able to virtually connect to the world. This generation’s culture may be more seamless in a globally connected environment [3]. This population segment was also unique in the context of the COVID-19 vaccination because it was categorized as a low-risk group, and they were required to wait for longer to be vaccinated.

Previous studies on COVID-19 vaccination intention or hesitancy among university students have been conducted in several countries, including the Czech Republic [4], Lebanon [5], and Bangladesh [6]. The results from multiple studies have shown that COVID-19 vaccination hesitancy does exist among university students, although they are not the majority [4,5]. It has been suggested that various sociodemographic factors such as degree major [6], nationality, residency status, university rank [5], experience with vaccinations [5], fear of side effects [6], a lack of knowledge and information, and a lack of trust regarding the COVID-19 vaccine would affect university students’ vaccination intentions [4,5,6].

These previous studies have identified the characteristics of university students that might affect vaccination intention but not actual behavior, and as single-country studies, they have not thoroughly examined the social and cultural aspects. 

### 1.3. Cultural Tightness-Looseness

Cultural tightness-looseness (CTL) is defined as the strength of social norms and the degree to which these are enforced within societies [7]. CTL is built on societal experiences of ecological and human-made threats. Such threats strengthen norms and eliminate deviant behavior to better coincide with society so that people can survive threatening situations such as natural disasters, terrorist attacks, and pandemics [8,9]. The more these challenges nations face, the stronger they build order and social coordination by developing strong norms [10].

Glenfield et al., based on a thorough review of previous research, have proposed a measurement that assesses the degree to which social norms are pervasive, clearly defined, and reliably imposed within nations. It consists of six items responded by individuals, and its reliability and validity have been established [11]. 

Considering the population of university students, the impact of social norm perception on behavioral decisions has been pointed out. One of the most famous and effective interventions was the alcohol abuse intervention among college students, and it has been widely used across the US [12]. The approach which addressed the gap between how much alcohol they thought other students were drinking and the actual consumption which was much smaller, found that alcohol overconsumption or unhealthy drinking behavior could be reduced. The huge success of such interventions suggests the importance of social norm perception of CTL among university students for vaccination behavior. 

#### CTL in the Context of COVID-19

Gelfand et al. examined how CTL was associated with countries’ successes in limiting COVID-19 cases and deaths by October 2020 [10]. They hypothesized that tight cultures with strict norms and punishments for deviance would have fewer cases and death rates than loose cultures with less stringent norms and estimated the relationship between CTL and COVID-19 cases and mortality rates among 33 countries as of 16 October 2020. After the under-reporting of demographics, geopolitical factors, other cultural dimensions, and climate were controlled for, the results showed that compared with culturally tight nations, culturally loose nations tended to have had 4.99 times the number of cases and 8.71 times the death rates. Thus, tightening social norms may confer an evolutionary advantage in times of a collective threat. 

Gelfand’s study provides solid evidence of CTL in the context of the COVID-19 pandemic, where national-level policies needed to be adequate to control behaviors. However, they did not examine whether CTL played any roles in people’s preventive behaviors, including vaccination, which could lead to limiting COVID-19 cases and deaths. 

Focusing on vaccination, Ng and Tan assessed whether CTL played a role in vaccination acceptance [13]. They examined publicly available data on the global attitude toward COVID-19 vaccination, in which respondents from 15 countries were surveyed between 30 December 2020 and 11 January 2021 regarding their willingness to receive the COVID-19 vaccine [14]. Using the data of 12 countries with available CTL scores (out of 15 surveyed countries), they examined the relationship between each country’s vaccination willingness and their CTL level, as defined by Gelfand et al. [11]. The result was unexpected as CTL was negatively related to the willingness to receive the COVID-19 vaccine, contradicting the theory proposed by Gelfand et al. [7], which suggested that societies with tighter cultures would compel individuals to cooperate more in crises. They pointed out that shared concerns about the side effects and safety profile of the new vaccine [15] and a low risk of contracting COVID-19 were perceived by people in countries with a well-controlled COVID-19 situation, which may have lowered the willingness to get vaccinated. They suggested that the positive effects of CTL in controlling COVID-19 cases and fatalities may work as a confounding factor in COVID-19-vaccination willingness and acceptance. 

Schmidt-Petri et al. assessed how individual preventive attitudes and behaviors toward COVID-19 were shaped by four groups of covariates: individual sociodemographics, health, personality, and regional-level controls [16]. They analyzed real-time representative survey data and quantified the extent to which differences in the averages of the covariates between two tight-culture countries, Germany and Japan, explained the differences in the observed preventive attitudes and behaviors. In Germany, a higher rate responded to the pandemic through preventive behaviors (e.g., handwashing), while in Japan, a higher percentage responded by being vaccinated. Attitudes and behaviors varied more according to individual characteristics in Germany than in Japan. 

Their results indicated that Japan’s lower infection and fatality rates compared to Germany were due to stronger norms, which suppressed individual differences such as socioeconomic status or individual personality traits and made people more willing to be vaccinated. However, they did not support the results of Gelfand’s study (2021), which revealed a relationship between the CTL level and infection outcomes [10].

Schumpe et al. assessed whether individual perceptions of COVID-19, the community, and the government would predict adherence to preventive behaviors to reduce the spread of the virus [17]. Using longitudinal data from an international panel, they analyzed whether CTL could predict a list of attitudes and behaviors to prevent the virus from spreading. The measurements included infection risk and conspiracy beliefs for the individual perception, social norms on distancing and community punishment for the perception about the community, and trust in the governmental response and communications about COVID-19 for the perception about the government. They also analyzed changes in several preventive behaviors, such as washing hands, avoiding crowds, quarantining, meeting friends and family in person, and meeting others in person; the only behavior that CLT predicted was leaving the house (measured by the number of days in a week). CTL did not predict changes in support for mandatory vaccination while an individual’s trust in the government to fight COVID-19, social distancing norms, and a right-wing political identity did. No difference was found among countries. 

The study results by Schumpe et al. ([17]) repeatedly indicated no CTL relationship between vaccination and other preventive behaviors. Their study included individual social and cultural perceptions other than CTL, such as trust in the government, political stance, and social norms on distancing, which indicated a relationship with vaccination behavior [17]. A mechanism might exist between CTL and such sociocultural-psychological factors, but it would vary by various preventive behaviors.

### 1.4. Study Aims

Previous studies have identified the characteristics of university students that might affect vaccination intention but not actual behavior. These are single-country studies that have not thoroughly examined the social and cultural aspects concerning vaccination, including the unique cultural concepts in the era of a highly globalized and virtually connected world with the information technology that this generation has always known. It is worthwhile to learn how to approach vaccination behavior for future pandemics from the perspective of university students. 

Previous research has demonstrated the critical role of CTL; however, its mechanism of action remains relatively unknown. CTL may play different roles in different behaviors [17] and social and cultural contexts [16], and it has not shown a relationship with vaccination behavior; however, so far, the primary focus has been on intention rather than behavior.

It is necessary to consider what is needed for the next step to understand better how CTL, as one of the sociocultural factors influencing people’s behaviors, could work for COVID-19 vaccination among youth. Thus, this study aims to examine whether CTL is related to COVID-19 vaccination behavior among university students across countries. 

The results of this study provide practitioners with strategic ideas for approaching the young generation regarding vaccination among important public health issues. They also provide the CTL theory body with additional insights, mainly adding evidence of the young generation, whose perceptions of culture might have changed from those of the older generation.

## 2. Methods

### 2.1. Participants

This study included students from India, Japan, and the US. These countries were chosen not only because of the researchers’ base countries, but also because of their differences in CTL. India is categorized as tight, the US as loose, and Japan is in the middle [11]. In addition, the infection rates and vaccination implementation processes differed among the three countries, each being unique. 

The participants in this study were selected from the survey panels of Ipsos, a global research company aligned with research companies in many countries (https://www.ipsos.com, accessed on 6 November 2023). Survey participants were recruited evenly, considering their geographical locations, so that the sample could cover each country. 

A total of 500 university students from each country participated. The screening questions were designed to include only undergraduate students who majored in any academic area, except those with health-related majors such as medicine, nursing and social work, considering that pursuing health-related careers would affect vaccination behavior. 

Data were collected from 500 applicable students in each country between 9 and 15 September 2022. This period was the shortest in Japan (until 10 September) and the longest in the US (until 15 September). Participants received incentives through various points that they could use in each country.

### 2.2. Measurements

#### 2.2.1. Language

Questions were asked in English in the US and India and in Japanese in Japan. The question items were first listed in the document in Japanese. The questions that originated in Japan and were translated into English were highlighted in the entire question list for the authors from the US (JM) and India (YA) to check them with more attention. If the item was difficult to understand or there were any problems in expression, such as misused words, the native authors and YK discussed the proper expression that should be consistent with the original meaning in Japanese. If the original scale was available in either language, it was used without translation. Although it is out of this study’s scope, the only item originated from another country, digital health literacy, was validated by another study [18].

#### 2.2.2. Dependent Variable: Vaccine Behavior

The questionnaire asked about the first, second, and third vaccinations, with response options “I got it”, “I plan to get”, “I don’t have a plan to get”, and “I don’t know.”

The progress of vaccination policy implementations varied in each country, and the rates of at least one dose were 84.0% in Japan, 79.2% in the US, and 72.3% in India as of 1 September 2022 [19]. To consider the experience of vaccine side effects, the answer to the second shot was used to assess vaccination behavior; thus, respondents were excluded from the analyses if they did not receive the first. In the analyses, the answers “I got it” and “I plan to get it” were considered “vaccinated (1)”, and others were “not vaccinated (0)”.

#### 2.2.3. Independent Variables

The independent variables included the respondents’ nation (India, Japan, or the US), CTL, and other cultural factors focusing on the individuals’ peripheral environmental feelings.

##### Cultural Tightness-Looseness

The scale is available in multiple languages and has been used in international studies [17], and this study used the English and Japanese versions. The scale asks the respondents to rate the extent to which they agree by choosing from six levels: (1) *strongly disagree*; (2) *moderately disagree*; (3) *slightly disagree*; (4) *slightly agree*; (5) *moderately agree*; and (6) *strongly agree*. The participants responded to the following six statements: (1) people are supposed to live by many social norms in this country; (2) in this country, expectations for how people should act in most situations are very clear; (3) people agree on what behaviors are appropriate versus inappropriate in most situations; (4) people in this country have a great deal of freedom in how they want to behave in most situations; (5) in this country, if someone acts in an inappropriate way, others will strongly disapprove; and (6) people in this country almost always comply with social norms. Item (4) was reverse recorded.

Cronbach’s α was calculated for six items and five items, excluding Item (4), which were 0.526 and 0.781, respectively. Considering the relatively low α for the six items and suggestions from previous research to remove it [20], we removed Item (4). The average score of the five items was calculated and used in the analyses.

##### Other Sociocultural Factors

Variables reflecting the respondents’ perceptions of the social environment other than CTL were considered. These variables included trust in the government, perceptions of vaccines, and vaccine behavior, which are COVID-19 context-specific. The variables focused on the peripheral and environmental feelings that individuals may perceive.

1.Trust in government

Trust in the government has been a critical factor affecting vaccine acceptance and preventive behaviors in the COVID-19 situation [17] where political measures are in power. Respondents were asked, “In general, how much do you trust the government of your country to take the right measures to deal with the COVID-19 pandemic?” They chose their responses from *not trustworthy at all* (1), *not very trustworthy* (2), *cannot say either* (3), *trustworthy* (4), or *a great deal* (5). This variable was dichotomized and 1–3 were coded as “0,” and 4 and 5 were coded as “1.”

2.Perceptions of vaccines and vaccination behavior

Respondents were asked how much they agreed with the statements “I believe receiving the COVID-19 vaccination is my societal responsibility” and “I received the COVID-19 vaccine because everyone around me was vaccinated,” along with others related to COVID-19 vaccination. The two questions assessed social responsibility and peer pressure, which are perceived expectations from the social environment or relationships with surrounding individuals considered proximate to CTL but distinctive, respectively. The answer was chosen from four options ranging from *strongly disagree* (1) to *strongly agree* (4). These were dichotomized and 1–2 were coded as “0,” and 3 and 4 were coded as “1.”

3.Control factors

Side effects

Previous studies conducted among university students have revealed that the fear of side effects was related to COVID-19 vaccination hesitancy [6]. This study asked if respondents experienced side effects at each vaccination and used the first experience. Response choices “No,” “Almost none,” and “Yes, but they did not interfere with my life much” were combined. “Yes, they interfered with my life quite a bit” was the only side effect experience (1), and the combined three were no experience (0).

Experience of infections

Experiences of themselves or family members infected with COVID-19 might have affected vaccination behavior and were thus considered in the analyses. For their own experience, respondents were asked if they had been infected with the virus and chose from *never been infected* (1), *infected before vaccination* (2), *infected after the first vaccination* (3), *infected after the second vaccination* (4), *or infected after the third vaccination* (5). Since this study focused on the second shot, the experiences after the second and third shots would not affect it and thus were also considered as no experience; thus, (1), (4), and (5) were recorded as “0” (no past infection experience), and 2 and 3 were recorded as “1” (past infection experience). Regarding the experience of family members’ infection, the participants were asked if they had any family members who were infected, and they responded “yes” (1) or “no” (0).

Demographics

According to a previous study [6], a student’s major is related to COVID-19 vaccination intention; it is assumed to be related to differences in what students are exposed to in their social circles. This study excluded health-related majors such as medicine, nursing, and social work to ensure that the data were not biased by students with a more favorable attitude toward COVID-19 vaccination. Respondents chose from “Humanities” (humanities, social sciences, etc.), “Sciences” (natural sciences, life sciences, etc., excluding medical and welfare sciences),” and “Interdisciplinary” for their majors.

In addition to the study major, gender (female, male, and others/do not want to tell), year in university (first, second, third, and fourth and more), and use of public transportation to go to university (yes or no) were used for analyses. Public transportation use, which indicates university students’ lifestyles and socioenvironmental differences, might increase risk perception; therefore, it was included in this study.

Finally, underlying health issues may also affect vaccination behavior. Respondents were asked if they had any underlying health issues and chose “yes” (1) or “no” (0).

### 2.3. Analyses

Descriptive statistics were calculated. Differences in independent variables among the three countries were assessed by comparing means through analysis of variance (ANOVA), Kruskal–Wallis test, or distributions by chi-square analyses. Logistic regression analyses were used to determine whether CTL was related to COVID-19 vaccination behavior among university students, considering other sociocultural perceptions about vaccination across countries. With the data of only those who had received the first vaccination, logistic regression analyses were conducted with each independent and control variable in the first step. Then, the model including those with second shot status was identified by stepwise logistic regression according to the Akaike’s Information Criterion (AIC) levels, which began with a complete list of independent and control variables. After analyzing all the data, the same procedures were followed for each of the three countries.

The statistical software EZR [20] was used for analyses.

## 3. Results

### 3.1. Descriptive Statistics

Table 1 shows the frequencies of variables for all the data (*n* = 1500) and for each country (*n* = 500 each).

Overall, the number of years in university was equally distributed, but significant differences existed between the three countries (*p* < 0.001). For study majors, about 40% were humanities and sciences, and about 15% were interdisciplinary, while Japan had more humanities majors and India more science majors. Thus, the distribution differences between the three countries were statistically significant (*p* < 0.001). The overall gender distribution was almost even between women and men, but a statistically significant difference was observed among countries (*p* < 0.05). Overall, approximately 72% of the respondents used public transportation systems, while fewer in the US (61.8%, *n* = 309) than in Japan (77.8%, *n* = 389) and India (76.0%, *n* = 380), which was significantly different (*p* < 0.001). Approximately 12% reported having underlying health conditions overall—9% in Japan, 12% in the US, and 13% in India (*p* = ns).

Regarding the COVID-19 vaccine, the overall rate of the first shot was 85.6%. While the rates were close to 90% in India (88.4%) and in Japan (88.0%), the US rate was 81.4%, which represented a significant difference among the three countries (*p* < 0.001). The same trend was observed for the second: more in Japan (87.4%) and India (82.6%) and fewer in the US (69.2%). The rates of the three countries were statistically significant (*p* < 0.001). About 53% percent experienced side effects to some degree during the first vaccination, and approximately 10% reported a severe experience. Comparing the three countries, fewer respondents tended to experience side effects in the US (29%) and in India (55.6%) compared to Japan (74.8%; *p* < 0.001). About 22% had been infected with COVID-19 at any time so far overall, but more were infected in the US (29%) and fewer in Japan (18.8%) and India (18.6%; *p* < 0.001). Paralleling their own infection experiences, the experience of family members’ infection was in the same trend: more in the US (41.6%) and fewer in India (36.4%) and Japan (32.4%; *p* < 0.05).

### 3.2. Logistic Regression

The analyses for the second shot were conducted using the data of 1289 respondents who were vaccinated first.

### 3.3. Description of Independent Variables

The overall CTL’s mean of five items out of the original CTL measure [17], composed of six items, was 3.95 (SD = 0.92). The higher the scores, the tighter the perception of social norms, and the highest was the US (4.13, SD = 0.96), followed by Japan (3.99, SD = 0.71), and India (3.74, SD = 1.03). Due to the unequal distributions among the three countries (*p* < 0.001), the difference in their means was confirmed by the Kruskal–Wallis test (*p* < 0.001). Differences were identified in each pair of countries by the Steel–Dwass analysis (three pairs, Japan and the US, Japan and India, and the US and India (*p* < 0.001; Table 2).

Table 3 shows the distribution of responses to other sociocultural perceptions, trust in the government, perception of vaccines as a social responsibility, and perception of others’ vaccination behavior. All variables were identified to be statistically significantly different (*p* < 0.001).

Overall, only approximately 10% of respondents trusted their governments regarding measures for the COVID-19 situation. Approximately 81% considered vaccination a social responsibility; about 74% got vaccinated because others did. These sociocultural perception variables differed significantly in their distribution between the three countries (all factor variables, *p* < 0.001). Considering the differences in the three countries, Japanese respondents were less likely to trust the government than Americans and Indians (“trust” responses: 1.82% in Japan, 13.02% in the US, and 15.32% in India). Japanese respondents were also less likely to consider vaccination a social responsibility (“agree” responses: 65.23% in Japan, 84.28% in the US, and 92.76% in India); however, they were more likely to get vaccinated because others did (“agree” responses: 80.68% in Japan, 71.50% in the US, and 70.14% in India). Japan consistently differed significantly from the US and India (*p* < 0.01), which, in turn, differed significantly only in their perception of vaccines as a social responsibility (*p* < 0.001).

### 3.4. Second Shot, Cultural Looseness-Tightness, and Other Sociocultural Behavior

#### 3.4.1. The Entire Sample (Total *n* = 1289)

Table 4 summarizes the results from a series of logistic regressions for the total.

The variables shown to be statistically significant in the univariate analyses were the CTL average, experience of infection, country, and study major (*p* < 0.05). Additionally, the use of public transportation was marginally significant (*p* < 0.1). The odds ratio of the CTL average was 1.55 (95% confidence interval [CI] 1.02–2.37), meaning that as the CTL average increased by 1 point, the likelihood of respondents getting the second shot increased by 55%. If respondents experienced infection before or after the first vaccination, their chance of getting the second shot decreased by 69% (OR = 0.31, 95% CI 0.21–0.79). The US respondents were 85% less likely to get a second shot than the Japanese respondents (OR = 0.15, 95% CI 0.03–0.68). For respondents majoring in interdisciplinary study areas, they were 79% less likely to get a second shot than those in the humanities (OR = 0.21, 95% CI 0.20–0.64).

Considering the model with all the variables, the CTL average and study majors turned marginally significant, while the infection experience stayed significant (OR = 0.32, 95% CI 0.10–0.99, *p* < 0.05). In addition, country as a variable turned marginally significant, although the US respondents had a lower likelihood of getting the second shot than the Japanese (OR = 0.15, 95% CI 0.03–0.82, *p* < 0.05).

The final model was obtained by the stepwise modeling method with the AIC, while the AIC for the model with all the variables was 237.41; the final model was 222.76, which included the CTL average (*p* < 0.05), the experience of infection (*p* < 0.05), the experience of family members’ infection (*p* < 0.1), country (*p* < 0.1), and study major (*p* = ns). The result showed that the chance of getting the second shot would be increased by 59% as the CTL average increased by 1 point, considering other factor variables in the model (OR = 1.59, 95% CI 1.04–2.44, *p* < 0.05). Respondents who experienced infection before or after the first vaccination were 72% less likely to receive the second shot, controlling for other factor variables in the model (OR = 0.28, 95% CI 0.09–0.85). Although the country was marginally significant as a variable, respondents from the US were 81% less likely to get the second shot than those from Japan (OR = 0.19, 95% CI 0.04–0.89), which was statistically significant (*p* < 0.05). Similarly, the study major was not significant as a variable; however, those majoring in interdisciplinary study areas were 81% less likely to get the second shot than those in humanities, which was statistically significant (OR = 0.29, 95% CI 0.09–0.93, *p* < 0.05).

#### 3.4.2. Three Countries

Table 5 presents the results of each country’s univariate and stepwise logistic regression analyses.

A significant variable from the univariate analyses was only the CTL average for Japan, and the result showed that as the CTL average increased by 1 point, the likelihood of getting the second shot would increase by 5.4 times (OR = 5.40, 95% CI 1.25–23.30, *p* < 0.05). The final model for Japan, with an AIC score of 20.21, included the CTL average (*p* < 0.05), perception of vaccination as a social responsibility (*p* = ns), and experience of family members’ infection (*p* = ns). As the CTL average increased by 1 point, the likelihood of getting the second shot would increase 56.3 times (OR = 56.3, 95% CI 1.14–2790), controlling for the factor variables in the model.

For the US, the results from the univariate analyses indicated that the CTL average and experience of infection were significant variables (*p* < 0.05). As the CTL average increased by 1 point, the likelihood of getting the second shot would increase by 89% (OR = 1.89, 95% CI 1.06–3.37). Those who experienced infection before or after the first vaccination were 72% less likely to get the second shot than those who did not (OR = 0.28, 95% CI 0.08–0.98). The study major was marginally significant as those majoring in interdisciplinary study areas were significantly less likely to get the second shot than those in humanities (OR = 0.16, 95% CI 0.03–0.83, *p* < 0.05).

The final model from the stepwise logistic regression analysis for the US included the CTL average and experience of infection (*p* < 0.05), of which the AIC was 105.9. The increase of 1 point in the CTL average would increase the chance of getting the second shot by about two times (OR = 2.02, 95% CI 1.10–3.69), while those with the experience of infection before or after the first vaccination were 75% less likely to get the second shot (OR = 0.25, 95% CI 0.07–0.90), controlling for each other.

The univariate analyses Identified no significant variables with regard to India. The final model with an AIC score of 87.36 included the perception of vaccination as a social responsibility (*p* < 0.1), experience of infection (*p* < 0.05), experience of family members’ infection (*p* < 0.05), and public transportation use (*p* = ns). Those who experienced infection before or after the first vaccination were 87% less likely to get the second shot (OR = 0.13, 95% CI 0.02–0.96), while those without the experience of family members’ infection were 91% less likely to get the second shot (OR = 0.09, 95% CI 0.01–0.99), controlling for other factor variables in the model.

## 4. Discussion

### 4.1. Cultural Looseness-Tighness, Vaccination Behavior, and Sociocultural Factors

Table 6 summarizes the logistic regression analysis results to examine how CTL was related to vaccination behavior, considering the sociocultural perceptions of the COVID-19 vaccine.

#### 4.1.1. Consistent Cultural Looseness-Tightness Roles but Differences in Manifestation among Countries

The results revealed a relationship between CTL scores and vaccination behavior. The higher the CTL average, the higher the likelihood of respondents getting a second shot. Other sociocultural perceptions, including trust in government, which was suggested for its relationship with vaccination behavior or perception in previous research [17], were excluded from the statistical modeling process. In addition, other factors that suggested their importance for vaccination among university students, such as side effects and study majors [6], were not included in the model. These results indicated that they did not explain vaccination behavior, as the univariate analyses showed the same results.

Other variables shown to be significant in explaining vaccination behavior were country, experience with infection, and study major. American and Indian students were less likely to receive a second shot, whereas other conditions stayed the same. While the result of India revealed that CTL was not in the model to explain vaccination behavior, CTL appears to play different roles depending on the context. Thus, we can incorporate it into health communication strategies for vaccination or other behaviors recommended during emergent situations, such as the COVID-19 pandemic. However, it is also important to consider the effectiveness of this method.

They were less likely to receive a second shot if they had been previously infected, which indicates that students considered their risk of another infection or severe symptoms to be reduced owing to the infection experience. Their self-efficacy may have been developed by overcoming their infection symptoms after the experience, which might have led them to not get vaccinated. Although the authorities do not necessarily recommend a series of COVID-19 vaccinations, booster vaccinations were publicly recommended at the time of the survey when the pandemic was not completely resolved. Publicly available information is diverse in terms of content and delivery. While public health authorities try to provide individuals with information that is highly complex and scientific and contains under-discussed facts, as the COVID-19 situation was, it is particularly challenging to deliver what is genuinely needed to each individual because of the diversity in how individuals perceive the publicly provided information and the experiences they might have had when infected.

Interdisciplinary majors were less likely to receive a second shot than humanities majors. This is consistent with previous research reporting a relationship between major and vaccination hesitancy [6]. Choosing what to study (more personal traits) and having been exposed to such an environment in constructing their perceptions may be related. These findings offer valuable insights into how to tailor approaches to university students by considering their demographic characteristics.

#### 4.1.2. CTL in General

The results of this study confirmed the importance of CTL in COVID-19 vaccine behavior among university students. Previous research [13,16,17] has examined how individuals thought about COVID-19 vaccination but not whether they took it; its impact, however, appeared different between countries. In India, which was previously categorized as having a high CTL level [11], CTL seemed to play no role in individuals receiving the second shot. Generally, CTL was less important in India than in the US and Japan.

The order of the CTL average scores was unexpected; considering previous research results [11] the US should have been lower than India and Japan. The reason might be the sampling, which calls for attention to CTL variations in countries, such as areas of living, age, and other demographic variables. Random sampling with a good sample size to secure representatives will be needed if we want to create a CTL index to compare countries reliably and validly.

For CTL measurement, we used the methods described in a previous study [11]. The results for the total (all three countries) tended to be parallel to those for the US, while the results for India and Japan tended to be inconsistent. We used the English and Japanese versions in India and Japan, respectively, for which the translation was validated. As expected, CTL measurement may not have been performed. The respondents in this study were young, and the measurement may not have been culturally sensitive enough in its expression. This should be examined in future studies.

### 4.2. Comparison of Three Countries

#### 4.2.1. Japan

Thus, CTL appears to be important for vaccination in Japan;however, it was unique in having shallow trust in the government for COVID-19 political measurements and perceived social responsibility for COVID-19 vaccination compared to the US and India. On the other hand, Japanese students tended to perceive others’ vaccinations as their motivation to get vaccinated, compared to Americans and Indians.

These results indicate that Japanese students refer to others and collective thoughts to identify what they are expected to think about matters. It is said that this tendency was carried across all personal characteristics, considering that no demographics were included in any model, which is consistent with the results of a previous study [16].

In Japan, “reading the air” is necessary and might represent the results. CTL, or perceived social norms, might be rooted in the culture and shared deeply in Japan, although the CTL average was lower than that in the US.

As a public strategy in Japan, promoting particular behaviors by “everyone is doing, so you should do it” might be more effective than “not enough are doing, so you should do it,” which is currently more common. This is consistent with the successful intervention approach to university students’ alcohol abuse prevention [12].

#### 4.2.2. US

Interestingly, although the US has been considered one of the culturally loosest countries, this study’s data showed that it was the tightest. The US was paralleled with the total data tendency, showing a consistent relationship between CLT, vaccination behavior, and sociocultural perceptions related to COVID-19. CTL would play a role, as hypothesized in the US, in enhancing vaccination behavior, which is a pro-social and other socially positive perception. These observations might be because American student respondents received the CTL measurement better than respondents from Japan and India, where the translated or English versions were used. However, the measurement did not originate from the native languages of Japanese and Indian respondents.

How can CTL be utilized for vaccination behavior in the US? CTL is useful when considering the characteristics of the groups being approached. Unlike Japan, individual ideas are diverse in the US. However, possible segments may be identified from the standpoint of CTL application in health communication strategies for issues such as COVID-19.

#### 4.2.3. India

According to the results of this study, CTL does not play a helpful role in enhancing behavior and perception related to vaccination. Diverse ideas seem to exist among generations, and public health must strategize health communication activities for vaccination or emergent issues, such as the COVID-19 situation in India. Media usage and information sources on COVID-19 were outside the scope of this study. For future research, it will be helpful to obtain insights into the kinds of information channels Indian university students used, how they understood and perceived the information, and what actions they took. Answers to these questions will allow us to approach Indian youth from a practical, real-life standpoint, which seems more appropriate for the Indian case, in which CTL appears insignificant.

### 4.3. Limitations and Future Research

Although the geographic areas of each country were considered for the study sample acquired from survey panels, the sample size was 500 from each country, which is not sufficient to universalize the study results. In addition to the sample size, registering with the survey panels might be an individual characteristic; thus, they may not be true population representatives. These results should be interpreted with caution. For instance, the respondents might have been keener on the COVID-19 vaccination issue, and thus, the results might have been skewed compared to possible ones of the general population.

This study revealed how different the three countries are and what their characteristics are. International comparisons are helpful for us to capture the entire picture by putting us in the position where objective standards are established and used for the comparison.

This study pointed out the discrepancies in CTL rankings among countries, which calls for a deeper understanding of CTL variations. Future research should consider developing a CTL index for cross-country comparisons, emphasizing robust random sampling. This might relate to the cultural insensitivity of the CTL measure in translation, highlighting the importance of further investigation in linguistically diverse contexts.

## 5. Conclusions

In summary, our study underpins the pivotal role of CTL in shaping individual decisions regarding COVID-19 vaccination. CTL’s influence surpassed that of individual sociocultural perceptions of the pandemic, which emphasizes its significance. However, differences across the three countries studied—particularly India—revealed that the impact of CTL is context-dependent.

Analyzing the results from each country offers valuable insights for future public health communication strategies during pandemic scenarios. CTL appeared consistently beneficial in Japan, independent of other demographic variables. In the US, it remained essential but exhibited variable effectiveness depending on individual characteristics, necessitating tailored approaches. Conversely, CTL did not appear to be influential in India, which suggests the need to explore alternative factors for effective communication.

Despite such limitations, mainly those related to sampling, this study enriches our understanding of CTL’s role in the young generations’ COVID-19 vaccination behavior and underpins its relevance in emergency situations such as the COVID-19 pandemic, highlighting the need for continued research in this field.

## Figures and Tables

**Table 1 vaccines-11-01821-t001:** Descriptive statistics of all respondents (*n* = 1500).

	All (*n* = 1500)		Japan (*n* = 500)		US (*n* = 500)		India (*n* = 500)	
	*n*	%	*n*	%	*n*	%	*n*	%
Country								
Japan	500	33.3	500	100.0				
US	500	33.3			500	100.0		
India	500	33.3					500	100.0
Year in university ***								
1st year at university	326	21.7	100	20.0	120	24.0	106	21.2
2nd year at university	423	28.2	*100*	20.0	162	32.4	161	32.2
3rd year at university	407	27.1	109	21.8	139	27.8	**159**	31.8
More than 4 years at university	344	22.9	**191**	38.2	79	15.8	74	14.8
Study major ***								
Humanistes (humanistes, social sciences, etc.)	650	43.3	**314**	62.8	217	43.4	*119*	23.8
Sciences (natural sciences, life sciences, etc. excluding medical and welfare sciences)	629	41.9	157	31.4	193	38.6	**279**	55.8
Interdisciplinary	221	14.7	*29*	5.8	90	18.0	102	20.4
Sex *								
Female (1)	727	48.5	243	48.6	250	50.0	234	46.8
Male (2)	747	49.8	247	49.4	237	47.4	263	52.6
Other (3)	18	1.2	6	1.2	12	2.4	0	0.0
Don’t want to answer (3)	8	0.5	4	0.8	1	0.2	*3*	0.6
Use of public transportation to university ***								
Yes (1)	1078	71.9	389	77.8	*309*	61.8	380	76.0
No (0)	422	28.1	111	22.2	191	38.2	120	24.0
Underlying health conditions								
Yes (1)	173	11.5	47	9.4	59	11.8	67	13.4
No (0)	1327	88.5	453	90.6	441	88.2	433	86.6
1st vaccination status ***								
(1) I got it	1289	85.9	440	88.0	*407*	81.4	442	88.4
(2) I plan to get	64	4.3	8	1.6	**38**	7.6	18	3.6
(3) I don’t have plan to get	97	6.5	**45**	9.0	33	6.6	*19*	3.8
(4) I don’t know	50	3.3	*7*	1.4	22	4.4	21	4.2
2nd vaccination status ***								
(1) I got it (1)	1196	79.7	**437**	87.4	*346*	69.2	413	82.6
(2) I plan to get (1)	70	4.7	*1*	0.2	**49**	9.8	20	4.0
(3) I don’t have plan to get (0)	19	1.3	*1*	0.2	**12**	2.4	6	1.2
(4) I don’t know (0)	4	0.3	1	0.2	0	0.0	3	0.6
N/A (those without 1st)	211	14.1	60	12.0	93	18.6	58	11.6
Side effects of 1st vaccination ***								
(1) No (0)	492	32.8	*66*	13.2	**262**	52.4	164	32.8
(2) Almost none (0)	274	18.3	**107**	21.4	*69*	13.8	98	19.6
(3) Yes, but they did not interfere with my life much (0)	374	24.9	**176**	35.2	*57*	11.4	141	28.2
(4) Yes, they interfered with my life quite a bit. (1)	149	9.9	**91**	18.2	19	3.8	39	7.8
Experience of infection ***								
(1) Never been infected (0)	1170	78.0	406	81.2	*355*	71.0	409	81.8
(2) Infected before vaccination (1)	147	9.8	*20*	4.0	62	12.4	65	13.0
(3) Infected after 1st vaccination (1)	38	2.5	*5*	1.0	**18**	3.6	15	3.0
(4) Infected after 2nd vaccination (0)	93	6.2	40	8.0	43	8.6	*10*	2.0
(5) Infected after 3rd vaccination (0)	52	3.5	29	5.8	22	4.4	*1*	0.2
Experience of family members’ infection *								
Yes (1)	552	36.8	*162*	32.4	208	41.6	182	36.4
No (0)	948	63.2	338	67.6	292	58.4	318	63.6

Notes: “*”: *p* < 0.05, “***”: *p* < 0.001, the numbers in “( )” are the recoded categories in the following analyses. *Italicized* numbers indicate statistically significantly small, while **bolded** ones are large (*p* < 0.05). If only one cell is shaded, it means that the country is statistically different from the other two, but the two others are not different. If two cells are shaded, two countries are statistically different, but each of two is not different from the other. If three cells are shaded, three countries are statistically different.

**Table 2 vaccines-11-01821-t002:** CTL mean scores for the total (*n* = 1289) and three countries.

Average Score of 5 Items (Excluding Item 4)	Total (*n* = 1289)		Japan (*n* = 440)		US (*n* = 407)		India (*n* = 442)	
	Mean	SD	Mean	SD	Mean	SD	Mean	SD
	3.95	0.92	3.99	0.71	4.13	0.96	3.74	1.03

Kruskal-Wallis chi-squared = 44.216, df = 2, *p*-value = 2.504 × 1/10^10^. Steel-Dwass Multiple Comparison: Japan:US t = 3.39 *, Japan:India t = 4.07 *, US:India t = 6.22 * (*: *p* < 0.001).

**Table 3 vaccines-11-01821-t003:** Distribution of responses to sociocultural factor variables for the total (*n* = 1289) and three countries.

Other Social-Cultural Factors: Perceptions of Vaccines and Vaccination Behavior		Total (*n* = 1289)		Japan (*n* = 440)		US (*n* = 407)		India (*n* = 442)	
		*n*	%	*n*	%	*n*	%	*n*	%
Trust in government ***									
	No trust	1160	89.99	432	98.18	354	86.98	374	84.62
	Trust	129	10.01	*8*	1.82	53	13.02	**68**	15.38
Perception of social responsibility ***									
	Disagree	249	19.32	153	34.77	64	15.72	32	7.24
	Agree	1040	80.68	*287*	65.23	343	84.28	**410**	92.76
Perception of others’ behavior ***									
	Disagree	333	25.83	*85*	19.32	116	28.50	132	29.86
	Agree	956	74.17	**355**	80.68	291	71.50	310	70.14

Notes: “***”: *p* < 0.001. *Italicized* numbers indicate statistically significantly small, while **bolded** ones are large (*p* < 0.05). If only one cell is shaded, it means that the country is statistically different from the other two, but the two others are not different. If two cells are shaded, two countries are statistically different from each other, but each of the two is not different from the other. If three cells are shaded, three countries are statistically different.

**Table 4 vaccines-11-01821-t004:** Summary of logistic regression analyses for the total (*n* = 1289).

	Univariate				Multivariate All				Stepwise Final Model			
	OR	95% LCL	95% UCL		OR	95% LCL	95% UCL		OR	95% LCL	95% UCL	
AIC	-				237.41				222.76			
(Intercept)	-	-	-	-	77.00	4.40	1.35 × 10^3^	**	102.00	9.54	1.10 × 10^3^	***
CTL average	1.55	1.02	2.37	*	1.57	1.00	2.47	+	1.59	1.04	2.44	*
Trust in government (trust)	0.87	0.38	1.98	ns	0.94	0.37	2.39	ns	-	-	-	-
Vaccine as social responsibility (agree)	1.49	0.58	3.81	ns	2.29	0.74	7.08	ns	-	-	-	-
Others’vaccination (agree)	1.26	0.51	3.09	ns	0.91	0.33	2.47	ns	-	-	-	-
Side effects of 1st vaccine (yes)	1.38	0.32	5.94	ns	0.67	0.14	3.11	ns	-	-	-	-
Experience of infection (yes)	0.31	0.12	0.79	*	0.32	0.10	0.99	*	0.28	0.09	0.85	*
Experience of family members’ infection (no)	0.75	0.31	1.83	ns	0.38	0.13	1.10	+	0.42	0.15	1.16	+
Country	-	-	-	*				+	-	-	-	+
US	0.15	0.03	0.68	*	0.15	0.03	0.82	*	0.19	0.04	0.89	*
India	0.22	0.05	1.02	+	0.28	0.05	1.66	ns	0.40	0.08	1.98	ns
Year in university *1	-	-	-	ns	0.75			ns	-	-	-	-
2nd	0.74	0.22	2.56	ns	0.69	0.19	2.50	ns	-	-	-	-
3rd	0.59	0.18	1.97	ns	0.49	0.14	1.77	ns	-	-	-	-
4th and over	1.06	0.26	4.28	ns	0.71	0.16	3.11	ns	-	-	-	-
Sex	-	-	-	ns				ns	-	-	-	-
Male	1.67	0.72	3.89	ns	1.65	0.64	4.23	ns	-	-	-	-
Others	9.91 × 10^5^	0.00	Inf	ns	8.07 × 10^5^	0.00	Inf	ns	-	-	-	-
Study major *2	-	-	-	*				ns	-	-	-	ns
Sciences (natural sciences, life sciences, etc. excluding medical and welfare sciences)	0.49	0.17	1.45	ns	0.65	0.21	2.02	ns	0.59	0.19	1.78	ns
Interdisciplinary	0.21	0.07	0.64	**	0.32	0.09	1.09	+	0.29	0.09	0.93	*
No public transportation use (no)	0.46	0.20	1.06	+	0.61	0.24	1.54	ns	-	-	-	-
No underlying health conditions (no)	1.75	0.59	5.22	ns	1.50	0.46	4.91	ns	-	-	-	-

Notes: AIC (Akaike information criterion) is an estimator of prediction error and thereby relative quality of statistical models for a given set of data. Thus, there is no set standard number to suggest the good fit. CTL (Cultural tightness-looseness) average is the average of 5 items (out of the original 6) scoring from 1 to 6. “*”: *p* < 0.05, “**”: *p* < 0.01, “***”: *p* < 0.001, “+”: *p* < 0.1, “Inf”: “infinite”, “ns”: “non-significant”.

**Table 5 vaccines-11-01821-t005:** Summary of univariate and stepwise modeling logistic regression analyses for the three countries.

	Japan (*n* = 440)								US (*n* = 407)								India (*n* = 442)							
	Univariate				Stepwise Final Model *				Univariate				Stepwise Final Model				Univariate				Stepwise Final Model			
	OR	95% L	95% U		OR	95% L	95% U		OR	95% L	95% U		OR	95% L	95% U		OR	95% L	95% U		OR	95% L	95% U	
AIC	-				20.21				-				105.90				-				87.36			
(Intercept)	-	-	-	-	2.06 × 10^6^	0.00	Inf	ns					2.82	0.29	27.60	ns					185	10.20	3360	***
CTL average	5.40	1.25	23.30	*	56.30	1.14	2790	*	1.89	1.06	3.37	*	2.01	1.10	3.69	*	1.07	0.56	2.01	ns	-	-	-	-
Trust in government (trust)	0.31	0.02	5.02	ns	-	-	-	-	1.83	0.57	5.88	ns	-	-	-	-	0.98	0.24	3.99	ns	-	-	-	-
Vaccine as social responsibility (agree)	8.36 × 10^7^	0.00	Inf	ns	2.49 × 10^9^	0.00	Inf	ns	1.07	0.23	5.02	ns	-	-	-	-	3.84	0.76	19.30	ns	4.67	0.85	25.70	+
Others’ vaccination (agree)	4.21	0.26	68.10	ns		-	-	-	0.83	0.22	3.13	ns	-	-	-	-	1.18	0.29	4.78	ns	-	-	-	-
Side effects of 1st vaccine (yes)	0.26	0.02	4.17	ns	-	-	-	-	3.69 × 10^6^	0.00	Inf	ns	-	-	-	-	0.77	0.09	6.32	ns	-	-	-	-
Experience of infection (yes)	5.38 × 10^5^	0.00	Inf	ns	-	-	-	-	0.28	0.08	0.98	*	0.25	0.07	0.90	*	0.63	0.13	3.10	ns	0.13	0.02	0.96	*
Experience of family members’ infection (no)	6.39 × 10^−8^	0.00	Inf	ns	4.17 × 10^−11^	0.00	Inf	ns	1.45	0.46	4.59	ns	-	-	-	-	0.20	0.03	1.64	ns	0.09	0.01	0.99	*
Year in university *1	-	-	-	ns	-	-	-	-	-	-	-	ns	-	-	-	-	-	-	-	ns	-	-	-	-
2nd	7.34 × 10^7^	0.00	Inf	ns	-	-	-	-	1.01	0.22	4.61	ns	-	-	-	-	5.38 × 10^−8^	0.00	Inf	ns	-	-	-	-
3rd	7.34 × 10^7^	0.00	Inf	ns	-	-	-	-	0.90	0.18	4.58	ns	-	-	-	-	3.30 × 10^−8^	0.00	Inf	ns	-	-	-	-
4th and over	1.94	0.12	31.40	ns	-	-	-	-	0.995	0.162	6.11	ns	-	-	-	-	7.26 × 10^−8^	0.00	Inf	ns	-	-	-	-
Sex	-	-	-	ns	-	-	-	-	-	-	-	ns	-	-	-	-	-	-	-	ns	-	-	-	-
Male	0.99	0.06	15.90	ns	-	-	-	-	2.45	0.73	8.28	ns	-	-	-	-	1.30	0.35	4.92	ns	-	-	-	-
Others	5.33 × 10^5^	0.00	Inf	ns	-	-	-	-	1.97 × 10^6^	0.00	Inf	ns	-	-	-	-	3.71 × 10^5^	0.00	Inf	ns	-	-	-	-
Study major *2	-	-	-	ns	-	-	-	-	-	-	-	+	-	-	-	-	-	-	-	ns	-	-	-	-
Sciences (natural sciences, life sciences, etc. excluding medical and welfare sciences)	8.45 × 10^6^	0.00	Inf	ns	-	-	-	-	0.37	0.07	1.92	ns	-	-	-	-	0.97	0.19	5.08	ns	-	-	-	-
Interdisciplinary	0.10	0.01	1.61	ns	-	-	-	-	0.16	0.03	0.83	*	-	-	-	-	0.86	0.12	6.24	ns	-	-	-	-
No public transportation use (no)	1.35 × 10^7^	0.00	Inf	ns	-	-	-	-	0.52	0.16	1.64	ns	-	-	-	-	0.38	0.10	1.44	ns	0.31	0.08	1.28	ns
No underlying health conditions (no)	2.32 × 10^−7^	0.00	Inf	ns	-	-	-	-	2.59	0.68	9.93	ns	-	-	-	-	0.96	0.12	7.82	ns	-	-	-	-

Notes: AIC (Akaike information criterion) is an estimator of prediction error and, thereby, the relative quality of statistical models for a given set of data. Thus, there is no set standard number to suggest a good fit. CTL (Cultural tightness-looseness) average is the average of 5 items (out of the original 6), scoring from 1 to 6. “*”: *p* < 0.05, “***”: *p* < 0.001, “+”: *p* < 0.1, “Inf”: “infinite”, “ns”: “non-significant”.

**Table 6 vaccines-11-01821-t006:** Result summary.

	CTL	Other Variables Remaining the Model with Statistical Significance (*p* < 0.05)
Total	*** [Pos]**	Country + (US *) [Neg] Experience of infection (yes) * [Neg] Study major + (Interdisciplinary *) [Neg]
Japan	*** [Pos]**	None
US	*** [Pos]**	Experience of infection (yes) * [Neg]
India	Not stayed in the model	Experience of infection (yes) * [Neg] Experience of family members’ infection (no) * [Neg]

Notes: “*”: *p* < 0.05, “+”: *p* < 0.1. “Pos” for positive “Neg” for negative directions of effects in “20”. In India, CTL did not remain in the model.

## Data Availability

The data set this study analyzed is not currently publicly available due to the limited sharing setting when approval was obtained from the IRB, but it is available on request from the corresponding author.

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
