# Peer review of "Cultural Tightness-Looseness: How Was It Related to Vaccination Behavior among University Students in Japan, the United States, and India?"

_vaccines, 2023, doi:10.3390/vaccines11121821_

Round 1
Reviewer 1 Report
Comments and Suggestions for Authors
The topic of the study is interesting and important, but this manuscript needs some improvements.
Abstract:
· Line 16, please add the word “Cultural” to tightness-looseness (CTL)
· Line 23, provide a full form of AIC.
· The sample size is conflicting and confusing (n=1500 in Line 19) and (n=1289 in Line 24). I would drop the former one.
Methodology
· Line 192: Give examples of health-related majors
· Line 199-204: Discuss questionnaire translation and back-translation procedures in greater detail.
· Measurement: The measurement scale points are different for variables (5-point, 3-point, 2-point). Was any type of standardization (e.g., z-score) performed before they were entered for the analyses?
· Line 219: 2.2.4. Cultural tightness-looseness 219. This should be a sub-section of 2.2.3. Independent variables. Thus, consider numbering it as 2.2.3.1.
· Line 236: 2.2.5. Other sociocultural factors. Number it as 2.2.3.2
Line 545-550. The Limitations section is extremely brief. Please consider renaming the section “Limitations and future research”. Then, expand the discussion of limitations and provide agendas for future research.
Comments on the Quality of English LanguageOverall, the quality of the language is good.
Author Response
Thank you very much for your feedback and please refer to the attached document.

Reviewer 2 Report
Comments and Suggestions for Authors
INTRODUCTION
Cultural tightness-looseness (CTL) is a construct. Explain How Cultural tightness-looseness (CTL) is mesaured?
In the introduction make a reference to social normal interventios and substance abuse in college students.
Try to use the tenses coherently eg: .eg in the introduction lines 94-98 are in past, 99-101 in present, and 102-104 again in past tense.
Material and Methods
The paragraph on language 2.21 is confusing, please rewrite it to make it mor clear.
Please include as an annex or supplementary material the two versions of the scale in English and Japanese.
Please indicate what software was used for the statistican analysis (eg. R, SPSS, stata, etc)
Results
In table 1, and 3 ,find in which cells there are statistically significant differences . You can do that by
Calculating adjusted standardized residuals that can be easily computed with many statistical program (eg :R, Ibm SPSS, stata or SAS)
In table 2 do a posteriori contrast using Bonferroni method to determine between what groups are differences.
At the feet of table table 5, explain what does it mean AIC, and CTL. (a reader should be able to understand a table without reading the text of the paper). Also explain what means inf (I suppose it is infinite)
Author Response

(The authors gave the same response as above.)

Reviewer 3 Report
Comments and Suggestions for Authors
General comment
I do not feel competent nor comfortable commenting English proficiency of the authors, however, I did struggle with the language at times and I have an impression that certain parts of the text and/or concepts got lost in the translation or in the cutting/editing process due to the word limitations. I would strongly suggest authors rewrite the abstract (particularly the sampling part) and perhaps even reconsider the title. The manuscript would generally benefit from a thorough language revision.
Examples:
“We should take lessons from our experience, especially considering the dynamics among sociocultural and individual factors on vaccination behaviors by national comparisons in the COVID-19 global context.” Pg 2 - not clear.
“This generation is assumed to hold unique cultural concepts from a unique globalization perspective and to always have virtual connectedness through information technology.” Pg 3 - This is mentioned twice in this section - why is this of relevance for CTL, and this study? Please elaborate.
“However, their results do not indicate how CTL could lead to the outcome.”Pg3 - please elaborate on what kind of outcome.
“CTL did not predict changes in support for mandatory vaccination but predicted an individual’s trust in the government to fight COVID-19, social distancing norms, a right-wing political identity, and the survey date.”Pg.4 - not clear, particularly the part on the survey.
“A mechanism might exist between CTL and such sociocultural-psychological factors, but it would vary by behavior”. What behavior? Not clear.
Specific comments
While this important study relies on a cross-cultural sample and targets different youth groups, allowing for comparisons across countries, the main limitations concern the writing style, methodology, and presentation of the results. The introduction of the paper introduces clearly the issue and demonstrates its importance, justifying the need to focus on this specific population, in three different social environments. The rationale for focusing on this population (youth from low, high, and middle CTL countries) is well-argued and clear, other than being digital natives. Perhaps the theoretical grounding of the study, over and above the pandemic, as described in the past literature could be addressed in greater depth and with more reference to prior work and theories.
Please further address the issue of the relatively small non-probabilistic sample (with diverse sociodemographic profiles of participants) and how it may have impacted findings (besides Limitations).
The methods are generally clear, although the analytical procedure could be better explained in some parts. Why logical regression? Some other controls/confounding variables could have been identified?
Independent variable(s) are not well defined - e.g.” other cultural factors focusing on individuals’ peripheral environmental feelings.”
Instrumentation should be better presented - how many scales in total were used (How many items in total? Just those listed in the text? Developed by which authors? Back-translated?)
The main limitation of the paper is its slight lack of analytical clarity and the discussion regarding the implications of the findings both for the literature and socially. Some claims do not seem to be supported by the data. The discussion part has findings that were either not discussed at all or not showcased properly in the analytical section. - e.g. “Their self-efficacy may have helped them overcome their infection symptoms after the experience.”
The discussion is a place to bring forward the findings that were reported earlier and show why these findings were important, what kind of knowledge they contribute, and, how they can be used, both academically and at a policy level. In general, the discussion could go more in-depth. How do the findings inform more specific recommendations? Consider strengthening the discussion with more detail on the implications of the findings for the specific population and society in general.
Author Response
General comment
I do not feel competent nor comfortable commenting English proficiency of the authors, however, I did struggle with the language at times and I have an impression that certain parts of the text and/or concepts got lost in the translation or in the cutting/editing process due to the word limitations.
I would strongly suggest authors rewrite the abstract (particularly the sampling part) and perhaps even reconsider the title. The manuscript would generally benefit from a thorough language revision.
Thank you for your suggestions. Because the other reviewers, along with you, gave us some comments on the revisions in the expression, we hope revising the points by the points would increase readability. We worked on the parts you suggested for the revisions as “Examples” below.
Examples:
“We should take lessons from our experience, especially considering the dynamics among sociocultural and individual factors on vaccination behaviors by national comparisons in the COVID-19 global context.” Pg 2 - not clear.
Thank you for pointing out, and we revised the sentence as below.
We should learn what actions we should take to promote behaviors such as vaccination among the youth under an emergent situation like the COVID-19 pandemic. Such actions will differ among different cultural contexts, and comparing countries with different cultures will allow us to identify them.
“This generation is assumed to hold unique cultural concepts from a unique globalization perspective and to always have virtual connectedness through information technology.” Pg 3 - This is mentioned twice in this section - why is this of relevance for CTL, and this study? Please elaborate.
Thank you for pointing out, and this part is being discussed for the relevance of the sample population of the study in general. The relevance for CTL is discussed when we mention the aim of the study. However, as you suggested, this was repeatedly mentioned, and unnecessary. Thus, the last sentence was removed.
“However, their results do not indicate how CTL could lead to the outcome.”Pg3 - please elaborate on what kind of outcome.
Thank you for pointing out, and we revised the sentence as below.
However, they did not examine whether CTL played any roles in people’s taking preventive behaviors, including vaccination, to lead to limiting COVID-19 cases and deaths.
“CTL did not predict changes in support for mandatory vaccination but predicted an individual’s trust in the government to fight COVID-19, social distancing norms, a right-wing political identity, and the survey date.”Pg.4 - not clear, particularly the part on the survey.
Thank you for pointing out, and we revised the sentence as below.
CTL did not predict changes in support for mandatory vaccination while an individual’s trust in the government to fight COVID-19, social distancing norms, a right-wing political identity, and the survey date did.
“A mechanism might exist between CTL and such sociocultural-psychological factors, but it would vary by behavior”. What behavior? Not clear.
Thank you for pointing out, and it was changed to “various preventive behaviors” inserted as below.
A mechanism might exist between CTL and such sociocultural-psychological factors, but it would vary by various preventive behaviors.
Specific comments
While this important study relies on a cross-cultural sample and targets different youth groups, allowing for comparisons across countries, the main limitations concern the writing style, methodology, and presentation of the results. The introduction of the paper introduces clearly the issue and demonstrates its importance, justifying the need to focus on this specific population, in three different social environments. The rationale for focusing on this population (youth from low, high, and middle CTL countries) is well-argued and clear, other than being digital natives.
Perhaps the theoretical grounding of the study, over and above the pandemic, as described in the past literature could be addressed in greater depth and with more reference to prior work and theories.
Thank you very much for your suggestion.
At the beginning, we had much more theoretical discussion in the manuscript. However, while brushing it up, we decided to eliminate those parts, considering the logical flow to make the point clear and readability for the audience in terms of the volume. Considering one of the editors comments for us to reduce the portion in the background, please forgive us to choose not to add information on the theoretical background.
Please further address the issue of the relatively small non-probabilistic sample (with diverse sociodemographic profiles of participants) and how it may have impacted findings (besides Limitations).
Thank you for your suggestion, and it is truly important to mention how the audience should read the results. Although you may suggest we should touch such limitations and cautions besides the limitation part, we think it difficult to do so, but we added the sentence following the part mentioning the sample limitation. We hope that it clarifies the matter better at least to some degree.
In addition to the sample size, registering with the survey panels might be an individual characteristic; thus, they may not be true population representatives. These results should be interpreted with caution. For instance, the respondents might have been keener on the COVID-19 vaccination issue, and thus, the results might have been skewed compared to possible ones of the general population.
The methods are generally clear, although the analytical procedure could be better explained in some parts. Why logical regression? Some other controls/confounding variables could have been identified?
Thank you for the comment. The logistics regression was chosen to simplify the analysis. We decided to focus on the second shot to include the vaccine side effects. We set the outcome as simple as possible by following the criterion of yes to the second shot.
Regarding the other controls/confounding factors, after examining the previous research on the COVID-19 vaccination, including those with the outcomes of perceptions or attitudes (not behaviors), we selected those in our study. We think that those included in this study would seemingly cover the range of concepts. On the other hand, this study was under a larger study about digital health literacy on the COVID-19 vaccination– and the number of variables (questions), and we could not include all variables/concepts that might have been confounding/control factors.
Independent variable(s) are not well defined - e.g.” other cultural factors focusing on individuals’ peripheral environmental feelings.”
Thank you for your suggestion. It was confusing because of the mistake in the numbering. We consider other cultural factors as independent variables, so the numbering was corrected as it should be.
Instrumentation should be better presented - how many scales in total were used (How many items in total? Just those listed in the text? Developed by which authors? Back-translated?)
Thank you for the comment. They are thoroughly described in 2.2 Measurement section, including the items. However, we will prepare an appendix of survey instruments, as another reviewer also suggested. Regarding the translation issue, we added detail descriptions of the process in 2.2.1 Language section as below. (highlighted parts are changed)
Questions were asked in English in the US and India and Japanese in Japan. The question items were first listed in the document in Japanese. The questions originated in Japan were translated into English were highlighted in the entire question list for the authors from the US (JM) and India (YA) to check them with more attention. If the item was difficult to understand or any problems in expression, such as misused words, the native authors and YK discussed the proper expression that should be consistent with the original meaning in Japanese. If the original scale was available in either language, it was used without translation. Although it is out of this study’s scope, the only item originated from the other country, digital health literacy, was validated by another study[18].
The main limitation of the paper is its slight lack of analytical clarity and the discussion regarding the implications of the findings both for the literature and socially. Some claims do not seem to be supported by the data. The discussion part has findings that were either not discussed at all or not showcased properly in the analytical section. - e.g. “Their self-efficacy may have helped them overcome their infection symptoms after the experience.”
The discussion is a place to bring forward the findings that were reported earlier and show why these findings were important, what kind of knowledge they contribute, and, how they can be used, both academically and at a policy level. In general, the discussion could go more in-depth. How do the findings inform more specific recommendations? Consider strengthening the discussion with more detail on the implications of the findings for the specific population and society in general.
Thank you very much for your comments.
First of all, the sentence “Their self-efficacy may have helped them overcome their infection symptoms after the experience” did not well described as we tried to explain. So that part was corrected as below.
Their self-efficacy may have been developed by overcoming their infection symptoms after the experience, which might have led them to not get vaccinated.
Besides, please forgive us to argue that we discussed the implications we proposed by pointing out specific findings (s)/results(s). We discussed specific practical/political recommendations in the specific population (Japan, the USA, and India) and in general. We would very much appreciate if you could provide us with what should be added besides what was presented in the manuscript.
Thank you very much for your feedback and please refer to the attached document.
